# HIV and Hepatitis C Risk among Tajik Migrant Workers Who Inject Drugs in Moscow

**DOI:** 10.3390/ijerph20115937

**Published:** 2023-05-24

**Authors:** Mary Ellen Mackesy-Amiti, Judith A. Levy, Mahbatsho Bahromov, Jonbek Jonbekov, Casey M. Luc

**Affiliations:** 1School of Public Health, University of Illinois at Chicago, Chicago, IL 60612, USA; 2PRISMA Research Center, Dushanbe 734029, Tajikistan

**Keywords:** HIV, hepatitis C, risk behavior, injection drug use, peer networks, Tajik migrant worker

## Abstract

The human immunodeficiency virus (HIV) epidemic in Eastern Europe and Central Asia continues to grow with most infections occurring in high-risk groups including people who inject drugs and their sexual partners. Labor migrants from this region who inject drugs while in Russia are at especially high HIV risk. Male Tajik migrant workers who inject drugs in Moscow (N = 420) were interviewed prior to a randomized trial of the Migrants’ Approached Self-Learning Intervention in HIV/AIDS (MASLIHAT) peer-education HIV-prevention intervention. Participants were interviewed about their sex and drug use behavior and tested for HIV and hepatitis C (HCV) prior to the intervention. Only 17% had ever been tested for HIV. Over half of the men reported injecting with a previously used syringe in the past month, and substantial proportions reported risky sexual behavior. Prevalence rates of HIV (6.8%) and HCV (2.9%) were elevated, although lower than expected when compared to estimates of prevalence among people who inject drugs at the national level in Tajikistan. Risk behavior in diaspora varied across the men’s regional area of origin in Tajikistan and occupation in Moscow, with HIV prevalence rates highest among those working at the bazaars. Evidence-based prevention approaches and messaging that specifically address the drug- and sex-related risk behavior of migrants with varying backgrounds are needed.

## 1. Introduction

Significant progress has been made in addressing the human immunodeficiency virus (HIV) pandemic throughout much of the world with new infections declining globally by 32% from 2010 to 2021 [1]. In Eastern Europe and Central Asia (EECA), however, new infections increased by 48% in the same time period [2]. The Joint United Nations Programme on HIV/AIDS (UNAIDS) reported that, in 2015, more than 80% of the region’s new infections were in the Russian Federation [3]. The HIV prevalence rate among persons age 15 to 49 in Russia was estimated at 1.5% in 2021 [4].

Labor migration is a major contributor to the movement and exchange of HIV across country borders and populations including EECA [5,6,7]. In Russia, many migrant workers originate from Central Asian countries, including Tajikistan—a small country with comparatively lower HIV rates, high poverty, and an ongoing opioid epidemic [8,9,10]. The HIV prevalence rate among adults in Tajikistan was estimated as 0.2% in 2021 [11]. Tajikistan faces numerous challenges in calculating how close it has come to meeting its UNAIDS Fast-Track target—ending the HIV/AIDS epidemic by 2030. Almost one million Tajik citizens work outside the country at any given time, with Russia the major destination. Such high rates of migration make it difficult to estimate the total number of Tajiks in-country and throughout the world who are at risk for and living with HIV [12].

Although the HIV epidemic in the Russian Federation has generalized throughout its population [13], injection drug use remains an important risk factor accounting for about 40% of new infections in recent years [2,14]. While UNAIDS estimates that 7.2% of people who inject drugs (PWID) in the Eastern European region are living with HIV [15], Russian sources estimate that 28% of PWID in Russia are living with the virus [16]. Labor migrants who inject drugs while in Russia are at especially high risk for acquiring HIV due to social marginalization and lack of access to health care and prevention services [17]. Meanwhile, migrants who become infected with HIV while living in the Russian Federation are at risk of deportation [18]. They also face a difficult choice between losing their income by returning home for treatment or remaining in Russia without access to HIV medical care [19]. 

Migrants who are unknowingly HIV positive lack the HIV treatment that they need until advancing symptoms drive them to seek medical attention. An analysis of the data from the Tajikistan Ministry of Health surveillance system found that among migrants living with HIV, more time spent in Russia predicted late presentation [20]. Such late presentation in initiating antiretroviral therapy can result in higher mortality and morbidity for migrants than might have occurred if begun sooner [21] and hinders government efforts in low resource countries in reaching the global fast-track goal of 90% of people living with HIV receiving treatment [22]. In addition, migrants who become unknowingly infected in a destination country can unintentionally transmit the virus to their spouses and other sexual partners upon returning home [23]. 

To address the need for preventive interventions for this population, the investigators developed and pilot tested the Migrants’ Approached Self-Learning Intervention in HIV/AIDS for Tajiks (MASLIHAT) intervention model for reducing risky drug, alcohol, and sexual behavior among male Tajik migrants who inject drugs while living in Moscow. MASLIHAT is a network-based, peer-educator training intervention developed as a socio-cultural adaptation of the Self-Help in Eliminating Life-Threatening Diseases (SHIELD) model [24]. Pilot testing conducted in 2018 demonstrated promising results with significant declines in risk behavior over six months among both peer-educator participants (PEs) and their network members with whom they regularly interacted [25].

The investigators are currently conducting a cluster-randomized parallel-groups trial to test the efficacy of the MASLIHAT intervention versus a comparison condition. This paper reports the risk behavior and HIV and hepatitis C virus (HCV) prevalence rates obtained from the study’s baseline survey and biological testing conducted with the combined sample of PEs and their network members recruited for both arms of the trial prior to the intervention. This analysis contributes to a slim but highly concerning body of literature documenting HIV risk and acquisition among labor migrants in diaspora. While most labor migrant studies tend to concentrate primarily on sexual risk [6,26,27,28,29,30,31], these data address a significant gap in what is known about migrants who engage in high-risk drug injection behavior while away from their home country. The focus on HIV risk in Russia is highly relevant as the country is the major destination for labor migrants from the Central Asian countries of Tajikistan, Kyrgyzstan, Uzbekistan, and in recently lessening numbers from Kazakhstan [32,33]. Attempts to curb HIV among PWID globally will never succeed without understanding and subsequently addressing the factors that drive transmission among Central Asia’s people who inject drugs (PWID) at home and in other countries [34]. The findings on Tajik male migrants who inject drugs while living in Moscow will help to address the need for epidemiological data on HIV/HCV risk and prevalence among Tajik migrants in diaspora. 

## 2. Methods

Study procedures were reviewed and approved by the Institutional Review Boards of the University of Illinois, Chicago (Chicago, IL, USA), PRISMA Research Center in Tajikistan, and the Moscow Nongovernment Organization “Bridge to the future”.

### 2.1. Recruitment

From October 2021 to April 2022, 420 male Tajik migrants who inject drugs were interviewed prior to a randomized trial of the MASLIHAT peer-education intervention. The sample includes 140 men who were recruited to the study to be trained as peer educators (PEs), and 280 injection drug use (IDU) network members recruited by the PEs. The 140 PEs were recruited from 12 sites in Moscow: 2 Tajik diaspora organizations, 4 bazaars, and 6 construction work sites. To be eligible to participate as a PE assigned to either the MASLIHAT intervention or the comparison control condition, prospective participants needed to be a male Tajik migrant aged 18 or older, a current or former PWID, give informed consent, intending to reside in Moscow for the next 12 months to participate in their assigned intervention and follow-up data collection, and willing to recruit two male PWID to participate as IDU network members for baseline and follow-up interviewing but who would not participate in the MASLIHAT or control condition educational sessions or activities. Network members (n = 280) had to meet the same eligibility criteria as PEs but also: (1) have injected drugs at least once in the last 30 days; and (2) be someone whom the PE sees at least once a week to permit them to share intervention information and encourage possible normative and behavioral change within their social networks. Participants in both study conditions received the customary compensation in Moscow of $20.00 for their time and transportation costs in participating in intervention sessions (PEs only) and for being interviewed at baseline and follow-up (both PEs and network members).

### 2.2. Baseline Interviews

After giving informed consent, baseline interviews with PEs and network members were conducted at the PRISMA office in Moscow or at a private location of the participant’s choosing. Following the interview, participants were referred to the Moscow HIV Prevention Center (Moscow, Russia) to be tested for HIV and HCV. Anonymized test results were reported to study staff with only a group number to identify the recruitment site. Participants who were diagnosed HIV positive were referred for treatment through the Tajikistan AIDS Center or the Russian AIDS Center. Treatment for HCV was arranged through the Tajik Diaspora Organization. 

### 2.3. Measures

The structured baseline questionnaire collected information on sociodemographic characteristics, migration characteristics and community involvement, substance use prior to migration and in the past six months in Moscow, sexual risk behavior, injection risk behavior, and HIV-related knowledge and attitudes.

HIV knowledge was assessed with 13 items (Cronbach alpha = 0.91). Eight items assessed knowledge of HIV transmission and non-transmission routes with response options being “safe”, “unsafe”, or “not sure”. Five additional true-false items assessed HIV-related knowledge such as, “You can look at a person and tell if they are infected with HIV”, and “There is a cure for HIV”. Correct responses were summed for a possible total score of 13 with responses of “not sure” counted as incorrect. As a measure of HIV risk awareness, participants were asked to rate their likelihood of contracting HIV as “not at all likely”, “somewhat unlikely”, “somewhat likely”, or “very likely”. They similarly rated the likelihood that someone with whom they inject drugs has HIV and the likelihood that they have shared syringes with someone who has HIV.

Alcohol-use measures included frequency of alcohol use in the past month and the Alcohol Use Disorders Identification Test (AUDIT) assessment of hazardous alcohol use and alcohol dependence [35]. Participants were asked, “How many days in the past month have you used alcohol, including beer, wine, or vodka?” Heavy drinking was assessed with AUDIT question #3, “How often do you have 6 or more drinks on one occasion”, with responses on a 5-point scale from “never” to “daily or nearly daily”. These responses were dichotomized for analysis as “never or less than monthly” vs. “at least monthly”. 

Sexual risk behavior outcomes included having sex with a female sex worker, multiple female sex partners, and having sex without a condom with a female sex partner in the past month. Past month condom use with regular and casual sexual partners and with sex workers was assessed separately on a 4-point scale as “never”, “sometimes”, “often”, or “always”. The items were combined to create a binary measure of “any condomless sex” vs. “sex with condoms or no sexual activity” in the past month.

Recent syringe sharing was assessed in response to the question, “When was the last time that you used a needle to shoot drugs after someone else used it first?” with 6 options: never, more than 4 weeks ago, in the last 4 weeks, within the last week, yesterday, or today. Responses were dichotomized to create a binary measure of having or not having used a shared syringe within the past month. Participants also were asked about syringe-cleaning practices with the question, “When you have used a needle after someone else, how often did you clean the syringe with [bleach, alcohol, water only, soap and water, or nothing]” with response options on a 6-point scale from “never” to “always”. 

Area of residence in Tajikistan (see Figure 1) was assessed by asking participants to select from a list the area of Tajikistan in which they had resided before migrating to Moscow. The list contained Tajikistan’s 4 administrative regions (the provinces of Khatlon, Sughd, and Gorno-Badakhshan Autonomous Oblast (GBAO), and the Districts of Republican Subordination—in Russian referred to as Rayoni Respublicanskogo Podchinenie (RRP)), and Dushanbe (Tajikistan’s capital city). 

### 2.4. Analysis

To begin, the analysis compared the sociodemographic attributes of PEs with those of their network members to identify possible significant differences using linear regression and multinomial logit models with cluster-adjusted standard errors. Bivariate tests of association were conducted to explore the relationships between 3 types of risk behavior (regular vs. heavy alcohol use, syringe sharing in the past month, and condomless sex in the past month) and the sociodemographic characteristics of participants using population-averaged Poisson regression models with an exchangeable within-group correlation structure and robust standard error estimation to obtain risk ratios and 95% confidence intervals.

## 3. Results

Table 1 shows the baseline demographic characteristics of the PEs (N = 140) and network members (N = 280). The PEs and network members reported similar demographic profiles. Network members were on average one year younger than PEs (B = −1.07, 95% CI −1.69–−0.46) and more likely to be on their first trip (Wald χ^2^ = 7.49, *p* = 0.02). Most participants had traveled to Moscow for work at least once previously, and the typical length of stay so far for this trip was 2–3 years.

Of the 420 participants, only 17% (n = 71) had been tested for HIV prior to participating in the study with one reporting that he tested positive and 14 declining to disclose their results. Over half (58%) reported they had been previously tested for hepatitis C (HCV). Participants recruited through a diaspora organization were more likely to have been tested for HIV (31%) compared to those recruited from construction sites (18%) or bazaars (8%). When requested to be tested for the study, 7 declined (1.7%). Of the 413 tested, 28 (6.8%) were HIV positive and 12 (2.9%) were HCV positive. All 12 who tested positive for HCV also tested positive for HIV. Testing results by recruitment venue are shown in Figure 2. Participants recruited from the bazaar sites were more likely than participants from construction sites to test positive for HIV (13.3 vs. 5.2%) and HCV (7.1 vs. 1.4%). None of the participants recruited from the diaspora sites tested positive for either HIV or HCV.

HIV knowledge and awareness responses are shown in Table 2. The average HIV transmission knowledge score was 7.2 (SD 3.3), or 55% correct. Migrants employed in loading and food service in the bazaars had lower knowledge scores compared to those employed in construction (Wald χ^2^ = 17.33, *p* = 0.0017), while those who had tested previously for HIV irrespective of occupation had higher knowledge scores (b = 2.07, 95% CI 1.24–2.89). More than one-third of participants who reported they had tested HIV negative prior to enrollment in the study believed they had no risk of acquiring HIV and were not at all worried, while less than 10% felt their risk was high and worried a lot. 

Prevalence of high-risk alcohol use, sexual risk behaviors, and injection risk behaviors are presented in Table 3. A large proportion of the men (39%) drank alcohol more than once a week, and 21% reported heavy drinking (6 or more drinks at a time) at least once a month. Sexual risk behavior was common, with 42% reporting condomless sex in the past month. Over 50% reported injecting with a previously used syringe in the past month. Among these, few (7%) ever cleaned their syringes with bleach and 58% never cleaned their syringes with anything. 

Associations between risk behaviors and sociodemographic characteristics (risk ratios and 95% confidence intervals) are shown in Table 4. Migrants who came from the Dushanbe area had the lowest rates of heavy alcohol use and syringe sharing while those from GBAO had the highest rates. Migrants who had attended university but left without a degree and those who were divorced were more likely to drink heavily. Condomless sex was most prevalent among migrants from the RRP and least prevalent among those from the Sughd province. Men with a college or university degree and those who were married or previously married were less likely on average to engage in condomless sex. Education and marital status had no bearing on the likelihood of using shared syringes. Risk behavior varied by employment category with men employed in construction less likely to engage in heavy alcohol use but more likely to report syringe sharing compared to their counterparts employed in selling and food service in the bazaars. 

## 4. Discussion 

The study’s findings confirm that Tajik labor migrants who inject drugs are at considerable risk for HIV and HCV when living in Moscow, a high HIV/HCV-prevalence destination city. Most participants in the study reported using a shared syringe without first cleaning it, and a large proportion reported condomless sex, multiple sex partners, and sex with sex workers. Many also engaged in heavy alcohol use on a regular basis, a life-style practice that can encourage risky drug and sexual behavior [36].

Due to COVID-19 travel restrictions, most of the participants in the current study had been in Moscow for 2 years or more; fewer than 10% had been in Moscow less than one year as compared with the more than one third of the earlier pilot study sample [25]. Furthermore, participants in the current study were more likely to report being at least somewhat worried about HIV (59 vs. 40%). Perhaps the advent of COVID-19 during the interim since the pilot was conducted had increased public general awareness, even among Tajik migrants [32], of being personally vulnerable to serious illness, loss of work, and other adverse consequences as the result of all types of viral transmission, including HIV and HCV. 

The study’s findings that report the influence of demographic characteristics on sex and alcohol risk suggest that marital status and education tend to exert a positive effect on migrant risk behavior. Yet, education and marital status appear to have no bearing on the likelihood of sharing an uncleaned syringe. Possibly, irrespective of a participant’s demographic background, sharing a syringe, even if unclean, is the result of most migrants’ fear of being reported to authorities as a PWID when obtaining new or clean syringes from a local pharmacy or needle exchange. 

HIV prevalence rates were significantly higher among bazaar workers than those working in construction, while no cases were found among migrants recruited from the diaspora organizations. These differing results may be related to prior-testing exposure. Participants recruited from diaspora organizations and those employed in construction were more likely to have been among the 17% in the study who had previously tested for HIV. Meanwhile prior testing was associated with greater knowledge of HIV risk and prevention, which also may help to explain why these two recruitment groups exhibited lower prevalence rates. 

The area of origin in Tajikistan appeared to strongly influence the participants’ HIV/HCV risk behavior while in Moscow. Each of these areas embody a somewhat different economic, cultural, and geographical environment that, in turn, may contribute to the differences found in the HIV/HCV risk and prevention behavior among the migrants who originate from them. Unlike the urbanized and comparatively well-resourced city of Dushanbe, GBAO is the poorest and most remote area of the country [37]. With no major industry, its unemployment rates are exceedingly high, and its small population differs ethnically and linguistically from the rest of Tajikistan. Migrants from GBAO tend to stay longer at their destination point and return home less frequently due to the remoteness of the area. Staying in Russia for long periods of time without citizenship, these migrants have little access to health care and information. When compared with each other, such regional socio-economic and geographical differences may help in some way to explain why participants from Dushanbe reported the lowest rates of heavy alcohol use and syringe sharing while those who migrated from the GBAO province reported the highest. Data from the Tajikistan Ministry of Health surveillance system (2006–2019) also showed that of people living with HIV, migrants compared to non-migrants were more likely to be from GBAO [20]. Meanwhile, condomless sex was the most prevalent among migrants from the RRP and the least prevalent among those from the Sughd province. Sughd is the second largest province in Tajikistan and is the most industrialized and economically developed. With special economic development zones and good employment opportunities, this region has better health care provision including greater access to HIV counseling and testing facilities. 

Of the 420 men who participated in our study, only 17% (n = 71) had been tested for HIV prior to participating in the study. Such a low testing rate is cause for concern. Of the 413 men who agreed to be tested at the study’s baseline, 28 (6.8%) tested positive for HIV and 12 (2.9%) tested positive for HCV. Labor migrants in diaspora avoid HIV testing out of fear of being deported, losing their current employment, or encountering stigma and social discrimination if diagnosed as HIV positive. Testing for HCV, on the other hand, was common, but not universal. 

Despite overall high levels of risk behavior and low testing rates, HIV and HCV prevalence (6.8% HIV positive and 2.9% HCV positive) were relatively low among the study’s male PWID participants when compared to the estimated prevalence of HIV/HCV at the national level among PWID in Tajikistan (13.5% HIV and 24.9% HCV in 2011) [10] and in the Russian Federation (30% HIV, 69% HCV based on multiple estimates from 2009–2015) [38]. The disparity may be at least partly due to sex differences, as national estimates include both men and women. In many settings, women who inject drugs have a higher prevalence of HIV [39,40]. Furthermore, the higher HIV prevalence rate for PWID in Tajikistan possibly reflects a sizable number of male Tajik migrants who returned home after being deported from a host country for having tested HIV positive or who voluntarily returned home for HIV treatment after acquiring the virus and/or testing positive elsewhere, including in Russia. Meanwhile, international immigrants seeking work in Russia must prove their HIV-negative status as a condition of being granted a residence work permit. 

Although an unknown number of labor migrants enter Russia either illegally or by submitting a fake certificate of a negative HIV test [41], fear of deportation or other negative consequences of being found to have HIV while in Moscow may restrict the flow of HIV-positive Tajik men who leave their home country for temporary work. 

*Limitations*. The study’s participants were sampled through in-person recruitment at 10 occupational sites and by referral from 2 Tajik service networks. Consequently, the study’s results may not generalize to those migrants whom these methods failed to reach or who chose not to participate. Biological testing for HIV and HCV was conducted by the Moscow HIV Prevention Center, an organization separate from the study that reported the results without personal identifiers to the MASLIHAT research staff. This anonymity was necessitated by the migrants’ reluctance to agree to confidential testing if their results could be linked to their identity and personal information. This safeguard, however, hinders our ability to test for possible associations between a participant’s HIV status and personal characteristics such as HIV knowledge, Tajikistan area of origin, and risk behavior. Moreover, the duration of HIV and HCV infection among participants is unknown; some may have acquired the infection prior to migration. Furthermore, the sample consists solely of men. Reliable statistics on the number of Tajik women who migrate to Russia for work are not readily available, although it is known to be low when compared to men. Nonetheless, their growing numbers [42] suggest the need for research that also considers their HIV/HCV risk behavior and prevalence while in diaspora. Areas of foci could include whether or not culturally defined gender expectations, drug use, and employment in different economic sectors of the Russian Federation translate to differences in female risk behavior. Finally, our data are cross-sectional. The examination of possible causal pathways will have to wait until the study’s longitudinal data collected at four follow-up time points are available. 

## 5. Conclusions

The alarming rate of HIV/HCV risk behavior among Tajik migrants who inject drugs in Moscow, and likely in other destination cities throughout Russia, calls for heightened prevention strategies and programs designed to reach this vulnerable population. Tajik migrants who inject drugs are not homogenous, however, in their HIV/HCV risk behavior. Evidence-based prevention approaches and messaging that specifically address the drug- and sex-related risk behavior of migrants with differing socio-demographic backgrounds from different parts of Tajikistan, and occupied in different employment sectors within the destination city, are needed. 

## Figures and Tables

**Figure 1 ijerph-20-05937-f001:**
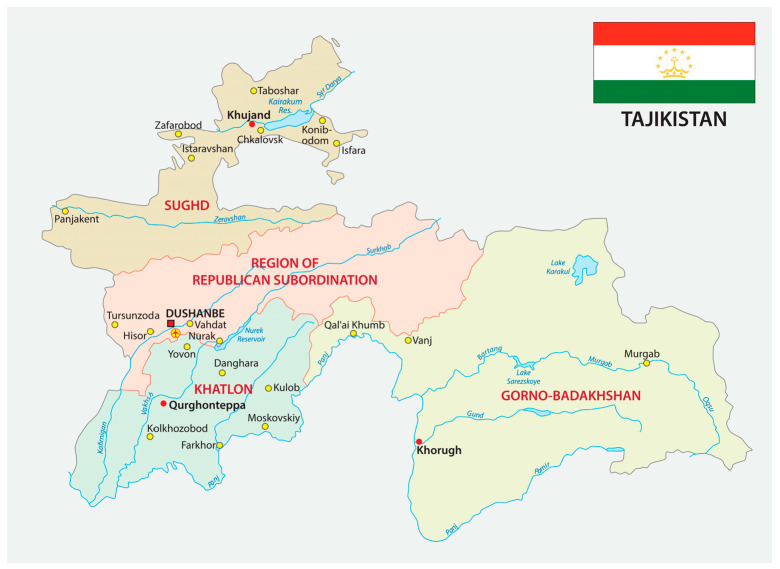
Map of Tajikistan.

**Figure 2 ijerph-20-05937-f002:**
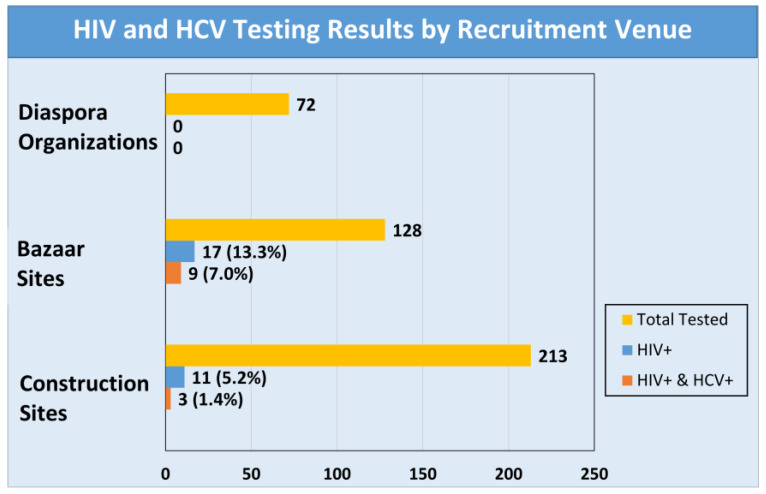
HIV and HCV test results by recruitment venue.

**Table 1 ijerph-20-05937-t001:** Demographic characteristics of PWID enrolled in the MASLIHAT trial.

	Peer Educator (n = 140) *		Network Members (n = 280) *		
Variable	Mean (SD)	Range	Mean (SD)	Range	*p*-Value ^†^
Age	30.7	21–50	29.6	19–49	0.001
	(6.74)		(5.90)		
	n	%	n	%	
Recruitment site					
Diaspora organization	24	17.1	48	17.1	
Bazaar	44	31.4	88	31.4	
Construction site	72	51.4	144	51.4	
Area of origin					0.280
Dushanbe	31	22.1	57	20.4	
Khatlon	29	20.7	59	21.1	
Sughd	13	9.3	31	11.1	
Gorno-Badakhshan	54	38.6	105	37.5	
Subordinate Districts	13	9.3	28	10.0	
Education					0.374
Secondary or less	91	65.0	165	58.9	
College or technical college	32	22.9	73	26.1	
University but no degree	6	4.3	8	2.9	
University degree	11	7.9	34	12.1	
Marital Status					0.200
not married	65	46.4	111	39.9	
married	14	10.0	38	13.7	
divorced	61	43.6	129	46.4	
How long in Russia this trip					0.280
One year or less	9	6.6	30	10.9	
>1 to 2 years	42	30.7	88	32.1	
>2 years	86	62.8	156	56.9	
How many trips to Moscow					0.024
One	8	5.7	41	14.6	
Two	55	39.3	81	28.9	
Three or more	77	55.0	158	56.4	
Employment					0.885
Construction	76	54.3	154	55.0	
Loading in bazaar	29	21.7	58	20.7	
Selling/food service	27	19.3	48	17.1	
Other/missing	8	5.7	20	7.1	

***** Study characteristics may not sum to column N due to missing information. ^†^ Linear regression for continuous and multinomial logit models for categorical variables with cluster-adjusted standard errors. PWID: people who inject drugs; MASLIHAT: Migrants’ Approached Self-Learning Intervention in HIV/AIDS.

**Table 2 ijerph-20-05937-t002:** HIV knowledge and awareness among PWID enrolled in the MASLIHAT trial (N = 420).

Variable	Median	IQR
HIV knowledge score	8.0	5–10
	n	%
How likely are you to get HIV		
Not at all likely	149	36.3
Somewhat unlikely	121	29.4
Somewhat likely	107	26.0
Very likely	34	8.3
How likely is it that at least one person you inject drugs with has HIV		
Not at all likely	150	36.5
Somewhat unlikely	146	35.5
Somewhat likely	85	20.7
Very likely	30	7.3
How likely is it that you have shared syringes with someone who has HIV		
Not at all likely	157	38.4
Somewhat unlikely	138	33.7
Somewhat likely	86	21.0
Very likely	28	6.9
How much do you worry about HIV		
Not at all	170	40.6
Somewhat	217	51.8
A lot	32	7.6

IQR: interquartile range. Note: Study characteristics may not sum to total N due to missing information. PWID: people who inject drugs; MASLIHAT: Migrants’ Approached Self-Learning Intervention in HIV/AIDS.

**Table 3 ijerph-20-05937-t003:** Risk behavior among PWID enrolled in the MASLIHAT trial (N = 420).

Variable	n	%
Frequency of alcohol use		
Monthly or less	70	16.7
2 to 4 times a month	187	44.6
2 to 3 times a week	142	33.9
4 or more times a week	20	4.8
Heavy alcohol use		
Never	218	52.2
Less than monthly	112	26.8
Monthly	72	17.2
Weekly	13	3.1
Daily or nearly everyday	3	0.7
Sex with sex worker in the past month	177	42.1
Multiple sex partners in the past month	125	29.8
Any condomless sex in the past month	178	42.4
Injected with shared syringe in the past month	228	54.3
Frequency of syringe sharing in past 3 months		
Never	101	24.1
Rarely	112	26.7
Less than half or half the time	90	21.4
About half the time	70	16.7
More than half the time	34	8.1
Almost always	11	2.6
Always	2	0.5
Cleaned syringe with bleach *		
Never	380	90.7
Once or twice/a few times	37	8.8
Sometimes/often	2	0.5
Cleaned syringe with water		
Never	265	63.2
Once or twice/a few times	131	31.3
Sometimes/often	23	5.5

Note: * Study characteristics may not sum to total N due to missing information. PWID: people who inject drugs; MASLIHAT: Migrants’ Approached Self-Learning Intervention in HIV/AIDS.

**Table 4 ijerph-20-05937-t004:** Associations between risk behaviors and sociodemographic characteristics of PWID enrolled in the MASLIHAT trial.

Variable	Heavy Alcohol Use (N = 418)		Syringe Sharing (N = 420)		Condomless Sex (N = 420)	
	PR	95% Conf. Int.	PR	95% Conf. Int.	PR	95% Conf. Int.
Age	1.02	(0.99, 1.05)	1.00	(0.98, 1.02)	0.96	(0.93, 0.98)
Recruitment site						
Diaspora organization	1.00		1.00		1.00	
Bazaar	0.98	(0.57, 1.67)	1.57	(0.99, 2.50)	0.83	(0.57, 1.20)
Construction site	0.71	(0.43, 1.15)	1.75	(1.12, 2.72)	0.86	(0.61, 1.21)
Area of Origin						
Dushanbe	1.00		1.00		1.00	
Khatlon	1.31	(0.66, 2.59)	1.48	(1.04, 2.10)	0.91	(0.61, 1.35)
Sughd	1.50	(0.69, 3.28)	1.54	(1.11, 2.15)	0.47	(0.28, 0.79)
Gorno-Badakhshan	2.06	(1.22, 3.49)	1.69	(1.25, 2.28)	1.23	(0.90, 1.67)
RRP	- ^†^		1.23	(0.77, 1.97)	1.82	(1.30, 2.53)
Education						
Secondary or less	1.00		1.00		1.00	
College or technical college	1.41	(0.94, 2.11)	0.93	(0.78, 1.11)	0.53	(0.38, 0.73)
University but no degree	2.38	(1.18, 4.79)	1.00	(0.61, 1.62)	0.55	(0.24, 1.26)
University degree	1.10	(0.58, 2.07)	0.99	(0.78, 1.27)	0.38	(0.20, 0.74)
Marital Status ^a^						
not married	1.00		1.00		1.00	
married	0.66	(0.32, 1.38)	1.25	(0.96, 1.63)	0.32	(0.18, 0.57)
divorced	1.60	(1.09, 2.33)	1.04	(0.83, 1.31)	0.55	(0.44, 0.69)
How many trips to Moscow						
one	1.00		1.00		1.00	
two	1.46	(0.72, 2.98)	0.76	(0.58, 1.01)	1.08	(0.76, 1.54)
three or more	1.59	(0.74, 3.41)	0.83	(0.64, 1.07)	0.72	(0.51, 1.04)
Employment						
Construction	1.00		1.00		1.00	
Loading in bazaar	1.12	(0.72, 1.74)	0.81	(0.60, 1.10)	1.20	(0.94, 1.54)
Selling/food service	1.79	(1.18, 1.74)	0.63	(0.44, 0.91)	0.72	(0.51, 1.01)
Other/missing	0.40	(0.12, 1.38)	1.31	(1.08, 1.58)	0.81	(0.44, 1.50)

^a^ Two missing. ^†^ Excluded due to non-varying outcome, N = 377. PWID: people who inject drugs; MASLIHAT: Migrants’ Approached Self-Learning Intervention in HIV/AIDS; RRP: Region of Republican Subordination.

## Data Availability

The datasets supporting the conclusions of this article are available in the Open Science Framework repository [Project ID: osf.io/7g3yh Data link: https://osf.io/ws5mp/ (accessed on 1 May 2023)].

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
