# Peer review of "HIV and Hepatitis C Risk among Tajik Migrant Workers Who Inject Drugs in Moscow"

_ijerph, 2023, doi:10.3390/ijerph20115937_

Round 1
Reviewer 1 Report
In the manuscript ijerph-2274770, several points need to be clarified, and some revisions are required as follows:
1. The abstract needs to be more representative of the study. For instance, the authors have not mentioned the name of the intervention model used i.e. the Migrants’ Approached Self-Learning Intervention in HIV/AIDS for 46 Tajiks (MASLIHAT) intervention model.
2. The situation in Tajikistan regarding migration and HIV should be clarified in the introduction. Hence, it is highly recommended to transfer the first paragraph in the discussion, “Lines 201-205,” to the introduction.
3. In all sections of the manuscript, the authors mentioned that 420 males were used. However, in the material and methods section, line 74, the authors mentioned that 140 males were recruited then in line 83, Network members (n=280). This could cause confusion to the reader. Hence, it is better to clarify in the first line of Recruitment that 420 males were used then to clarify the number of participants in both study conditions.
4. The discussion section needs to be deepened by comparing the results to similar earlier studies.
5. The use of abbreviations thought out the manuscript needs to be revised as many errors exist as follows:
- The authors should uniform the abbreviation for the disease, either HIV or AIDS.
- When using the abbreviation, the full term should be first mentioned, then the abbreviation further E.g. line 8, HIV should be presented as human immunodeficiency virus (HIV), then the abbreviation should be used further. The same for PWID in line 38; PWID should be people who inject drugs (PWID). Similar errors have been repeated many times throughout the manuscript.
- It is not preferable to begin the sentence with abbreviations like UNAIDs in line 28.
6. The writing style should be formal from the third-person perspective. Do not use we or our (E.g. line 58, “Our analysis contributes” should be “The current analysis contributes; line 141; we compared; line 143: we conducted)
Author Response
- The abstract needs to be more representative of the study. For instance, the authors have not mentioned the name of the intervention model used i.e. the Migrants’ Approached Self-Learning Intervention in HIV/AIDS for 46 Tajiks (MASLIHAT) intervention model.
Response: The abstract now mentions the name of the intervention model and it also spells out all acronyms that it mentions.
- The situation in Tajikistan regarding migration and HIV should be clarified in the introduction. Hence, it is highly recommended to transfer the first paragraph in the discussion, “Lines 201-205,” to the introduction.
Response: The paragraph (lines 201 -205) has been transferred to the introduction.
- In all sections of the manuscript, the authors mentioned that 420 males were used. However, in the material and methods section, line 74, the authors mentioned that 140 males were recruited then in line 83, Network members (n=280). This could cause confusion to the reader. Hence, it is better to clarify in the first line of Recruitment that 420 males were used then to clarify the number of participants in both study conditions.
Response: We have revised this section to clarify the study sample.
- The discussion section needs to be deepened by comparing the results to similar earlier studies.
Response: Few studies have been conducted on HIV risk among migrants in Russia irrespective of their country of origin, and most of these studies are at least 10 years old and were conducted by one or more of the authors of this paper. Their results are already reviewed in the paper.
- The use of abbreviations thought out the manuscript needs to be revised as many errors exist as follows:
- The authors should uniform the abbreviation for the disease, either HIV or AIDS;
Response: HIV is now used solely and uniformly throughout the text.
- When using the abbreviation, the full term should be first mentioned, then the abbreviation further E.g. line 8, HIV should be presented as human immunodeficiency virus (HIV), then the abbreviation should be used further. The same for PWID in line 38; PWID should be people who inject drugs (PWID). Similar errors have been repeated many times throughout the manuscript.
Response: All abbreviations in the text are now spelled out before being used elsewhere.
- It is not preferable to begin the sentence with abbreviations like UNAIDs in line 28.
Response: The sentence now begins with, “The Joint United Nations Programme on HIV/AIDS (UNAIDS) …”
- The writing style should be formal from the third-person perspective. Do not use we or our (E.g. line 58, “Our analysis contributes” should be “The current analysis contributes; line 141; we compared; line 143: we conducted)
Response: The pronouns “we” and “our” have been replaced with more formal grammatical terms and phrases throughout the text.
Reviewer 2 Report
This manuscript examines the risk of HIV-1 and to a lesser extent HCV among Tajik migrant workers who inject drugs while in Moscow. This work starts to track HIV-1 positive individuals from the central Asian country Tajikistan, and the likelihood that migrant workers who inject drugs will obtain HIV-1. The manuscript goes into great detail in trying to establish tendencies among the Tajik migrant workers by thoroughly examining their demographics and social/recreational tendencies, but fail to draw any unanimous conclusions, except for the fact that workers who inject drugs are more likely to obtain HIV-1. HCV is less well studied in this work and could be left out. However, it is important to make these conclusions and make aware the significance of HIV-1 transmission. A highlight of this work is the talented writing.
Author Response
Response: We thank the reviewer for calling attention to our underreporting of the study’s HCV results and have added additional findings about HCV to the text.
Reviewer 3 Report
Overview and general recommendation:
The authors aimed to evaluate the risk of HIV and HCV infections among Tajik male migrant workers who inject drugs in Moscow, Russia. The topic is interesting and important since this aspect is so far largely unknown. As a result, prevalence rates of HIV (6.8%) and HCV (2.9%) were lower than expected when compared to those among PWID at the national level in Tajikistan and in the Russian Federation, which warrants further investigation. In addition, risk behavior in the diaspora varied across the men’s regional area of origin in Tajikistan and occupation in Moscow, with HIV prevalence rates highest among those working at the bazaars. Despite unneglectable limitations, the study is overall well-designed, and the manuscript is well-written. I only have a few minor comments for the authors' consideration.
1. Curiously, HIV and HCV prevalence (6.8% HIV positive and 2.9% HCV positive) were relatively low among male PWID participants in this study, compared to the estimated prevalence of HIV/HCV at the national level among PWID in Tajikistan (13.5% HIV and 24.9% HCV in 2011) and in the Russian Federation (30% HIV, 69% HCV based on multiple estimates from 2009-2015). Could this relate to the recruitment conducted during COVID-19 from October 2021 to April 2022? In my opinion, there should be a huge sampling bias largely due to the fact that many HIV/HCV-positive migrants chose not to participate in this study.
2. Did the authors include the impact of the age of PWID on the risk of HIV and hepatitis C in this study? What about the education level?
3. To most people, Tajikistan is relatively unfamiliar. It would be highly beneficial for the readers to comprehend the locations of Dushanbe, Khatlon, Sughd, Gorno-Badakhshan, and Subordinate Districts if a map of Tajikistan could be provided. Otherwise, the authors can give a brief introduction to these Provinces at the beginning.
4. Is there any data available about the prevalence of HIV and HCV infection among migrant PWID from other countries in Moscow? If so, please discuss this.
Author Response
- Curiously, HIV and HCV prevalence (6.8% HIV positive and 2.9% HCV positive) were relatively low among male PWID participants in this study, compared to the estimated prevalence of HIV/HCV at the national level among PWID in Tajikistan (13.5% HIV and 24.9% HCV in 2011) and in the Russian Federation (30% HIV, 69% HCV based on multiple estimates from 2009-2015). Could this relate to the recruitment conducted during COVID-19 from October 2021 to April 2022? In my opinion, there should be a huge sampling bias largely due to the fact that many HIV/HCV-positive migrants chose not to participate in this study.
Response: As mentioned in the discussion (lines 304-308), the lower prevalence of HIV and HCV possibly reflects a sizable number of male Tajik migrants who return home for HIV treatment after acquiring the virus and/or testing positive in a destination country including Russia. Also, the text now explains that international immigrants seeking work in Russia must prove their HIV-negative status as a condition of being granted a residence work permit. Although an unknown number of labor migrants enter Russia either illegally or by submitting a fake-certificate of a negative HIV test result (see Kashnitsky, 2020), fear of deportation or other negative consequences of being found to have HIV while in Moscow may restrict the flow of HIV-positive Tajik men who leave their home country for temporary work. Additionally, the study’s focus on HIV risk and prevention was not revealed during recruitment so migrant PWIDs’ reluctance to participate in an HIV study was not at issue.
- Did the authors include the impact of the age of PWID on the risk of HIV and hepatitis C in this study? What about the education level?
Response: Age and education were included in the analysis of risk behavior (Table 4).
- To most people, Tajikistan is relatively unfamiliar. It would be highly beneficial for the readers to comprehend the locations of Dushanbe, Khatlon, Sughd, Gorno-Badakhshan, and Subordinate Districts if a map of Tajikistan could be provided. Otherwise, the authors can give a brief introduction to these Provinces at the beginning.
Response: A map of the area is now included.
- Is there any data available about the prevalence of HIV and HCV infection among migrant PWID from other countries in Moscow? If so, please discuss this.
Response: Given the threat of deportation for HIV and HCV infected migrants, most do not get tested in Russia. Therefore, these data are not available
Reviewer 4 Report
The article illustrates the prevalence and risk factors of HIV and HCV in Tajik migrant male workers to Moscow who inject drugs. This topic is important and the article is well-written. However, some comments need to be addressed.
1. In the abstract write the full word before the abbreviation 'PWID'.
2. In the introduction, the authors described the prevalence among PWID only in both countries. Please add a paragraph on the overall recent prevalence in Tajikstan and in Russia.
3. In the methods section, at Measures subheading: how were the psychosocial measures like the depression and migrant stigma done? and what was the results of these measures? and association with the disease?
4. In Measures subheading, the authors assessed the alcohol use. Alcohol is not a direct risk factor for either HIV or HCV. On the other hand, other risk factors such as previous operations, blood transfusion, history of partner with HIV were not evaluated.
5. In table 1 and table 2, the value of SD is not present in the table.
6. In results section, demonstrate the rate of previous HIV testing and the percentage of HIV mono and co-infection in a figure form instead of text.
7. In the assessment of frequency of syringe sharing past 3 months, the measure of 'More than half the time' is equal to 'Almost always'.. That represents confusion in the choice to the participants in the questionnaire. Consider them as one value instead of two.
8. In table 4, again remove the heavy alcohol use from the comparison because it is not a risk factor for HIV or HCV. Revise the number of those who have condomless sex (n=420) that represents the total number of participants.
9. In discussion, transfer this paragraph to the introduction from line 265-275
'Migrants who are unknowingly HIV positive lack the HIV treatment that they need until advancing symptoms drive them 266 to seek medical attention. An analysis of data from the Tajikistan Ministry of Health surveillance system found that among migrants living with HIV, more time spent in Russia predicted late presentation [31]. Such late presentation in initiating antiretroviral therapy can result in higher mortality and morbidity for migrants than might have occurred if begun sooner [32] and hinders government efforts in low resource countries to reaching the global fast-track goal of 90% of people living with HIV receiving treatment [33]. In addition, migrants who become unknowingly infected in a destination country can unintentionally transmit the virus to their spouses and other sexual partners upon returning home [34].'
10. In limitation of the study, add that the HIV and HCV status before enrollment in the study and before migration was not available, because the patient could have been infected before migration.
11. What was the impact of this study on those who were positive for either HIV and/or HCV? did they receive any treatment?
Author Response
1. In the abstract write the full word before the abbreviation 'PWID'.
Response: Done
2. In the introduction, the authors described the prevalence among PWID only in both countries. Please add a paragraph on the overall recent prevalence in Tajikistan and in Russia.
Response: We have added the requested prevalence rates.
3. In the methods section, at Measures subheading: how were the psychosocial measures like the depression and migrant stigma done? and what was the results of these measures? and association with the disease?
Response: We removed the text referring to these measures as they were not relevant to the analysis.
4. In Measures subheading, the authors assessed the alcohol use. Alcohol is not a direct risk factor for either HIV or HCV. On the other hand, other risk factors such as previous operations, blood transfusion, history of partner with HIV were not evaluated.
Response: It’s true that heavy alcohol use is not a direct factor for either HIV or HCV, but it was included as an outcome variable as it contributes to the likelihood of unsafe sex.
5. In table 1 and table 2, the value of SD is not present in the table.
Response: SD of age in Table 1 was in the wrong column. This has been corrected. Table 2 reports the median and interquartile range of knowledge scores.
6. In results section, demonstrate the rate of previous HIV testing and the percentage of HIV mono and co-infection in a figure form instead of text.
Response: We have added a figure to show the results of HIV and HCV testing.
7. In the assessment of frequency of syringe sharing past 3 months, the measure of 'More than half the time' is equal to 'Almost always'. That represents confusion in the choice to the participants in the questionnaire. Consider them as one value instead of two.
Response: Within the context of the scale, it is clear that “Almost always” is greater than “More than half the time.”
8. In table 4, again remove the heavy alcohol use from the comparison because it is not a risk factor for HIV or HCV. Revise the number of those who have condomless sex (n=420) that represents the total number of participants.
Response: As noted above, we include heavy alcohol use as an outcome as it potentially contributes to unsafe sex. The numbers in the column headings of the Table 4 indicates the number of participants included in the analysis (i.e. non-missing).
9. In discussion, transfer this paragraph to the introduction from line 265-275 ---
'Migrants who are unknowingly HIV positive lack the HIV treatment that they need until advancing symptoms drive them 266 to seek medical attention. An analysis of data from the Tajikistan Ministry of Health surveillance system found that among migrants living with HIV, more time spent in Russia predicted late presentation [31]. Such late presentation in initiating antiretroviral therapy can result in higher mortality and morbidity for migrants than might have occurred if begun sooner [32] and hinders government efforts in low resource countries to reaching the global fast-track goal of 90% of people living with HIV receiving treatment [33]. In addition, migrants who become unknowingly infected in a destination country can unintentionally transmit the virus to their spouses and other sexual partners upon returning home [34].'
Response: Done.
10. In limitation of the study, add that the HIV and HCV status before enrollment in the study and before migration was not available, because the patient could have been infected before migration.
Response: This possibility has been noted.
11. What was the impact of this study on those who were positive for either HIV and/or HCV? Did they receive any treatment?
Response: Participants who were diagnosed HIV positive were referred to treatment through the Tajikistan AIDS Center or the Russian AIDS Center. Most of those who tested HIV positive returned to Tajikistan for treatment. Others are receiving medication from Tajikistan through relatives, or they are receiving treatment from the Russian AIDS Center. For migrants remaining in Moscow, treatment for HCV was arranged through the Tajik Diaspora Organization.
Round 2
Reviewer 1 Report
The manuscript has significantly improved and is suitable for publication in its present form
Reviewer 4 Report
The authors replied to all comments and the illustrations and corrections were done appropriately. No more comments are required.